# EXPLICIT SPARSE TRANSFORMER: CONCENTRATED ATTENTION THROUGH EXPLICIT SELECTION

## ABSTRACT

Self-attention-based Transformer has demonstrated the state-of-the-art performances in a number of natural language processing tasks. Self-attention is able to model long-term dependencies, but it may suffer from the extraction of irrelevant information in the context. To tackle the problem, we propose a novel model called **Explicit Sparse Transformer**. Explicit Sparse Transformer is able to improve the concentration of attention on the global context through an explicit selection of the most relevant segments. Extensive experimental results on a series of natural language processing and computer vision tasks, including neural machine translation, image captioning, and language modeling, all demonstrate the advantages of Explicit Sparse Transformer in model performance. We also show that our proposed sparse attention method achieves comparable or better results than the previous sparse attention method, but significantly reduces training and testing time. For example, the inference speed is twice that of sparsemax in Transformer model.

## 1 INTRODUCTION

Understanding natural language requires the ability to pay attention to the most relevant information. For example, people tend to focus on the most relevant segments to search for the answers to their questions in mind during reading. However, retrieving problems may occur if irrelevant segments impose negative impacts on reading comprehension. Such distraction hinders the understanding process, which calls for an effective attention.

This principle is also applicable to the computation systems for natural language. Attention has been a vital component of the models for natural language understanding and natural language generation. Recently, Vaswani et al. (2017) proposed Transformer, a model based on the attention mechanism for Neural Machine Translation(NMT). Transformer has shown outstanding performance in natural language generation tasks. More recently, the success of BERT (Devlin et al., 2018) in natural language processing shows the great usefulness of both the attention mechanism and the framework of Transformer.

However, the attention in vanilla Transformer has a obvious drawback, as the Transformer assigns credits to all components of the context. This causes a lack of focus. As illustrated in Figure 1, the attention in vanilla Transformer assigns high credits to many irrelevant words, while in Explicit Sparse Transformer, it concentrates on the most relevant $k$ words. For the word "tim", the most related words should be "heart" and the immediate words. Yet the attention in vanilla Transformer does not focus on them but gives credits to some irrelevant words such as "him".

Recent works have studied applying sparse attention in Transformer model. However, they either add local attention constraints (Child et al., 2019) which break long term dependency or hurt the time efficiency (Martins & Astudillo, 2016). Inspired by Ke et al. (2018) which introduce sparse credit assignment to the LSTM model, we propose a novel model called **Explicit Sparse Transformer** which is equipped with our sparse attention mechanism. We implement an explicit selection method based on top-$k$ selection. Unlike vanilla Transformer, Explicit Sparse Transformer only pays attention to the $k$ most contributive states. Thus Explicit Sparse Transformer can perform more concentrated attention than vanilla Transformer.

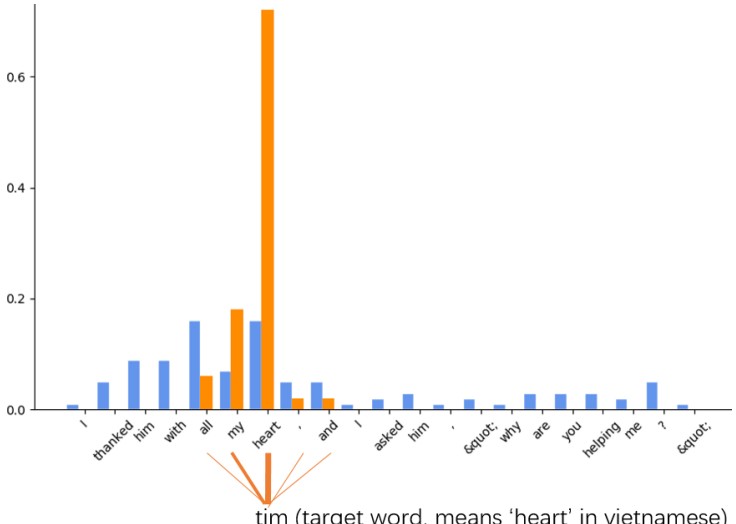

Figure 1: Illustration of self-attention in the models. The orange bar denotes the attention score of our proposed model while the blue bar denotes the attention scores of the vanilla Transformer. The orange line denotes the attention between the target word "tim" and the selected top-$k$ positions in the sequence. In the attention of vanilla Transformer, "tim" assigns too many non-zero attention scores to the irrelevant words. But for the proposal, the top-$k$ largest attention scores removes the distraction from irrelevant words and the attention becomes concentrated.

We first validate our methods on three tasks. For further investigation, we compare our methods with previous sparse attention methods and experimentally answer how to choose k in a series of qualitative analyses. We are surprised to find that the proposed sparse attention method can also help with training as a regularization method. Visual analysis shows that Explicit Sparse Transformer exhibits a higher potential in performing a high-quality alignment. The contributions of this paper are presented below:

- We propose a novel model called Explicit Sparse Transformer, which enhances the concentration of the Transformer's attention through explicit selection.

- We conducted extensive experiments on three natural language processing tasks, including Neural Machine Translation, Image Captioning and Language Modeling. Compared with vanilla Transformer, Explicit Sparse Transformer demonstrates better performances in the above three tasks. Specifically, our model reaches the state-of-the-art performances in the IWSLT 2015 English-to-Vietnamese translation.

- Compared to previous sparse attention methods for transformers, our methods are much faster in training and testing, and achieves better results.

## 2 PREMIERS

The review to the attention mechanism and the attention-based framework of Transformer can be found in Appendix A.1.

## 3 EXPLICIT SPARSE TRANSFORMER

Lack of concentration in the attention can lead to the failure of relevant information extraction. To this end, we propose a novel model, **Explicit Sparse Transformer**, which enables the focus on only a few elements through explicit selection. Compared with the conventional attention, no credit will be assigned to the value that is not highly correlated to the query. We provide a comparison between the attention of vanilla Transformer and that of Explicit Sparse Transformer in Figure 2.

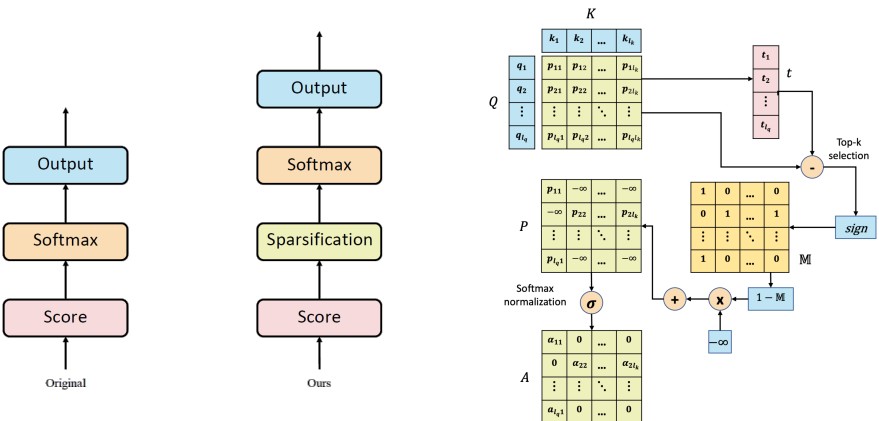

Figure 2: The comparison between the attentions of vanilla Transformer and Explicit Sparse Transformer and the illustration of the attention module of Explicit Sparse Transformer. With the mask based on top-$k$ selection and softmax function, only the most contributive elements are assigned with probabilities.

Explicit Sparse Transformer is still based on the Transformer framework. The difference is in the implementation of self-attention. The attention is degenerated to the sparse attention through top-$k$ selection. In this way, the most contributive components for attention are reserved and the other irrelevant information are removed. This selective method is effective in preserving important information and removing noise. The attention can be much more concentrated on the most contributive elements of value. In the following, we first introduce the sparsification in self-attention and then extend it to context attention.

In the unihead self-attention, the key components, the query $Q[l_Q, d]$, key $K[l_K, d]$ and value $V[l_V, d]$, are the linear transformation of the source context, namely the input of each layer, where $Q = W_Q x$, $K = W_K x$ and $V = W_V x$. Explicit Sparse Transformer first generates the attention scores $P$ as demonstrated below:

$$P = \frac{QK^{\mathrm{T}}}{\sqrt{d}} \tag{1}$$

Then the model evaluates the values of the scores $P$ based on the hypothesis that scores with larger values demonstrate higher relevance. The sparse attention masking operation $\mathcal{M}(\cdot)$ is implemented upon $P$ in order to select the top-$k$ contributive elements. Specifically, we select the $k$ largest element of each row in $P$ and record their positions in the position matrix $(i, j)$, where $k$ is a hyperparameter. To be specific, say the $k$-th largest value of row $i$ is $t_i$, if the value of the $j$-th component is larger than $t_i$, the position $(i, j)$ is recorded. We concatenate the threshold value of each row to form a vector $t = [t_1, t_2, \cdots, t_{l_Q}]$. The masking functions $\mathcal{M}(\cdot, \cdot)$ is illustrated as follows:

$$\mathcal{M}(P, k)_{ij} = \begin{cases} P_{ij} & \text{if } P_{ij} \geq t_i \text{ ($k$-th largest value of row $i$)} \\ -\infty & \text{if } P_{ij} < t_i \text{ ($k$-th largest value of row $i$)} \end{cases} \tag{2}$$

With the top-$k$ selection, the high attention scores are selected through an explicit way. This is different from dropout which randomly abandons the scores. Such explicit selection can not only guarantee the preservation of important components, but also simplify the model since $k$ is usually a small number such as 8, detailed analysis can be found in 5.2. The next step after top-$k$ selection is normalization:

$$A = \text{softmax}(\mathcal{M}(P, k)) \tag{3}$$

where $A$ refers to the normalized scores. As the scores that are smaller than the top k largest scores are assigned with negative infinity by the masking function $\mathcal{M}(\cdot, \cdot)$, their normalized scores, namely the probabilities, approximate 0. We show the back-propagation process of Top-k selection in A.3. The output representation of self-attention $C$ can be computed as below:

$$C = AV \tag{4}$$

| Model | En-De | En-Vi | De-En |
|---|---|---|---|
| ConvS2S (Gehring et al., 2017) | 25.2 | - | - |
| Actor-Critic (Bahdanau et al., 2017) | - | - | 28.5 |
| NPMT+LM (Huang et al., 2017) | - | 28.1 | 30.1 |
| SACT (Lin et al., 2018) | - | 29.1 | - |
| Var-Attn (Deng et al., 2018) | - | - | 33.7 |
| NP2MT Feng et al. (2018) | - | 30.6 | 31.7 |
| Transformer (Vaswani et al., 2017) | 28.4 | - | - |
| RNMT (Chen et al., 2018) | 28.5 | - | - |
| Fixup (Zhang et al., 2019) | 29.3 | - | 34.5 |
| Weighted Transformer (Ahmed et al., 2017) | 28.9 | - | - |
| Universal Transformer (Dehghani et al., 2018) | 28.9 | - | - |
| Layer-wise Coordination (He et al., 2018) | 29.1 | - | - |
| Transformer(relative position) (Shaw et al., 2018) | 29.2 | - | - |
| Transformer (Ott et al., 2018) | 29.3 | - | - |
| DynamicConv (Wu et al., 2019) | 29.7 | - | 35.2 |
| Local Joint Self-attention (Fonollosa et al., 2019) | 29.7 | - | 35.7 |
| Trasformer(impl.) | 29.1 | 30.6 | 35.3 |
| Explicit Sparse Transformer | **29.4** | **31.1** | **35.6** |

Table 1: Results on the En-De, En-Vi and De-En test sets. Compared with the baseline models, Explicit Sparse Transformer reaches improved performances, and it achieves the state-of-the-art performances in En-Vi and De-En.

The output is the expectation of the value following the sparsified distribution $A$. Following the distribution of the selected components, the attention in the Explicit Sparse Transformer model can obtain more focused attention. Also, such sparse attention can extend to context attention. Resembling but different from the self-attention mechanism, the $Q$ is no longer the linear transformation of the source context but the decoding states $s$. In the implementation, we replace $Q$ with $W_Q s$, where $W_Q$ is still learnable matrix.

In brief, the attention in our proposed Explicit Sparse Transformer sparsifies the attention weights. The attention can then become focused on the most contributive elements, and it is compatible to both self-attention and context attention. The simple implementation of this method is in the Appendix A.4.

## 4 RESULTS

We conducted a series of experiments on three natural language processing tasks, including neural machine translation, image captioning and language modeling. Detailed experimental settings are in Appendix A.2.

### 4.1 NEURAL MACHINE TRANSLATION

**Dataset** To evaluate the performance of Explicit Sparse Transformer in NMT, we conducted experiments on three NMT tasks, English-to-German translation (En-De) with a large dataset, English-to-Vietnamese (En-Vi) translation and German-to-English translation (De-En) with two datasets of medium size. For En-De, we trained Explicit Sparse Transformer on the standard dataset for WMT 2014 En-De translation. The dataset consists of around 4.5 million sentence pairs. The source and target languages share a vocabulary of 32K sub-word units. We used the *newstest 2013* for validation and the *newstest 2014* as our test set. We report the results on the test set.

For En-Vi, we trained our model on the dataset in IWSLT 2015 (Cettolo et al., 2014). The dataset consists of around 133K sentence pairs from translated TED talks. The vocabulary size for source language is around 17,200 and that for target language is around 7,800. We used *tst2012* for validation, and *tst2013* for testing and report the testing results. For De-En, we used the dataset in IWSLT 2014. The training set contains 160K sentence pairs and the validation set contains 7K sentences.

| Model | BLEU-4 | METEOR | CIDEr |
|---|---|---|---|
| SAT Bazzani et al. (2018b) | 28.2 | 24.8 | 92.3 |
| SCST Rennie et al. (2017) | 32.8 | 26.7 | 106.5 |
| NBT Lu et al. (2018) | 34.7 | 27.1 | 107.2 |
| AdaAtt Lu et al. (2017) | 33.2 | 26.6 | 108.5 |
| ARNN Bazzani et al. (2018a) | 33.9 | 27.6 | 109.8 |
| Transformer | 35.3 | 27.7 | 113.1 |
| UpDown Anderson et al. (2018) | **36.2** | 27.0 | 113.5 |
| Explicit Sparse Transformer | 35.7 | **28.0** | **113.8** |

Table 2: Results on the MSCOCO Karpathy test split.

Following Edunov et al. (2018), we used the same test set with around 7K sentences. The data were preprocessed with byte-pair encoding (Sennrich et al., 2016). The vocabulary size is 14,000.

**Result** Table 1 presents the results of the baselines and our Explicit Sparse Transformer on the three datasets. For En-De, Transformer-based models outperform the previous methods. Compared with the result of Transformer (Vaswani et al., 2017), Explicit Sparse Transformer reaches 29.4 in BLEU score evaluation, outperforming vanilla Transformer by 0.3 BLEU score. For En-Vi, vanilla Transformer[1] reaches 30.2, outperforming the state-of-the-art method (Huang et al., 2017). Our model, Explicit Sparse Transformer, achieves a new state-of-the-art performance, 31.1, by a margin of 0.5 over vanilla Transformer. For De-En, we demonstrate that Transformer-based models outperform the other baselines. Compared with Transformer, our Explicit Sparse Transformer reaches a better performance, 35.6. Its advantage is +0.3. To the best of our knowledge, Explicit Sparse Transformer reaches a top line performance on the dataset.

## 4.2 IMAGE CAPTIONING

**Dataset** We evaluated our approach on the image captioning task. Image captioning is a task that combines image understanding and language generation. We conducted experiments on the Microsoft COCO 2014 dataset (Chen et al., 2015a). It contains 123,287 images, each of which is paired 5 with descriptive sentences. We report the results and evaluate the image captioning model on the MSCOCO 2014 test set for image captioning. We used the publicly-available splits provided by Karpathy & Li (2015). The validation set and test set both contain 5,000 images.

**Result** Table 2 shows the results of the baseline models and Explicit Sparse Transformer on the COCO Karpathy test split. Transformer outperforms the mentioned baseline models. Explicit Sparse Transformer outperforms the implemented Transformer by +0.4 in terms of BLEU-4, +0.3 in terms of METEOR, +0.7 in terms of CIDEr. , which consistently proves its effectiveness in Image Captioning.

## 4.3 LANGUAGE MODELING

**Dataset** Enwiki8[2] is large-scale dataset for character-level language modeling. It contains 100M bytes of unprocessed Wikipedia texts. The inputs include Latin alphabets, non-Latin alphabets, XML markups and special characters. The vocabulary size 205 tokens, including one for unknown characters. We used the same preprocessing method following Chung et al. (2015). The training set contains 90M bytes of data, and the validation set and the test set contains 5M respectively.

**Result** Table 3 shows the results of the baseline models and Explicit Sparse Transformer-XL on the test set of enwiki8. Compared with the other strong baselines, Transformer-XL can reach a better performance, and Explicit Sparse Transformer outperforms Transformer-XL with an advantage.

---

[1] While we did not find the results of Transformer on En-Vi, we reimplemented our vanilla Transformer with the same setting.

[2] http://mattmahoney.net/dc/text.html

| Model | Params | BPC |
|---|---|---|
| LN HyperNetworks (Ha et al., 2016) | 27M | 1.34 |
| LN HM-LSTM (Chung et al., 2016) | 35M | 1.32 |
| RHN (Zilly et al., 2017) | 46M | 1.27 |
| Large FS-LSTM-4 (Mujika et al., 2017) | 47M | 1.25 |
| Large mLSTM (Krause et al., 2016) | 46M | 1.24 |
| Transformer (Al-Rfou et al., 2018) | 44M | 1.11 |
| Transformer-XL (Dai et al., 2019) | 41M | 1.06 |
| Adaptive-span (Sukhbaatar et al., 2019) | 39M | 1.02 |
| Explicit Sparse Transformer-XL | 41M | **1.05** |

Table 3: Comparison with state-of-the-art results on enwiki8. Explicit Sparse Transformer-XL refers to the Transformer with our sparsification method.

| Method | En-Vi | De-En | Training Speed (tokens/s) | Inference Speed (tokens/s) |
|---|---|---|---|---|
| Transformer | 30.6 | 35.3 | 49K | 7.0K |
| Sparsemax (Martins & Astudillo, 2016) | - | 31.2 | 39K | 3.0K |
| Entmax-1.5 (Peters et al., 2019) | 30.9 | 35.6 | 40K | 4.9K |
| Entmax-alpha (Correia et al., 2019) | - | 35.5 | 13K | 0.6K |
| Proposal | 31.1 | 35.6 | 48K | 6.6K |

Table 4: In the Transformer model, the proposed method, top-k selection before softmax is faster than previous sparse attention methods and is comparable in terms of BLEU scores.

## 5 DISCUSSION

In this section, we performed several analyses for further discussion of Explicit Sparse Transformer. First, we compare the proposed method of topk selection before softmax with previous sparse attention method including various variants of sparsemax (Martins & Astudillo, 2016; Correia et al., 2019; Peters et al., 2019). Second, we discuss about the selection of the value of $k$. Third, we demonstrate that the top-k sparse attention method helps training. In the end, we conducted a series of qualitative analyses to visualize proposed sparse attention in Transformer.

### 5.1 COMPARISON WITH OTHER SPARSE ATTENTION METHODS

We compare the performance and speed of our method with the previous sparse attention methods[3] on the basis of strong implemented transformer baseline. The training and inference speed are reported on the platform of Pytorch and IWSLT 2014 De-En translation dataset, the batch size for inference is set to 128 in terms of sentence and half precision training(FP-16) is applied.

As we can see from Table 4, the proposed sparse attention method achieve the comparable results as previous sparse attention methods, but the training and testing speed is 2x faster than sparsemax and 10x faster than Entmax-alpha during the inference. This is due to the fact that our method does not introduce too much computation for calculating sparse attention scores.

The other group of sparse attention methods of adding local attention constraints into attention (Child et al., 2019; Sukhbaatar et al., 2019), do not show performance on neural machine translation, so we do not compare them in Table 4.

---

[3]We borrow the implementation of Entmax1.5 in Tensorflow from `https://github.com/deep-spin/entmax`, and the implementation of Sparsemax, Entmax-1.5, Entmax-alpha in Pytorch from `https://gist.github.com/justheuristic/60167e77a95221586be315ae527c3cbd`. We have not found a reliable Tensorflow implementation of sparsemax and entmax-alpha in the transformer (we tried to apply the official implementation of sparsemax in Tensorflow to tensor2tensor, but it reports loss of NaN.)

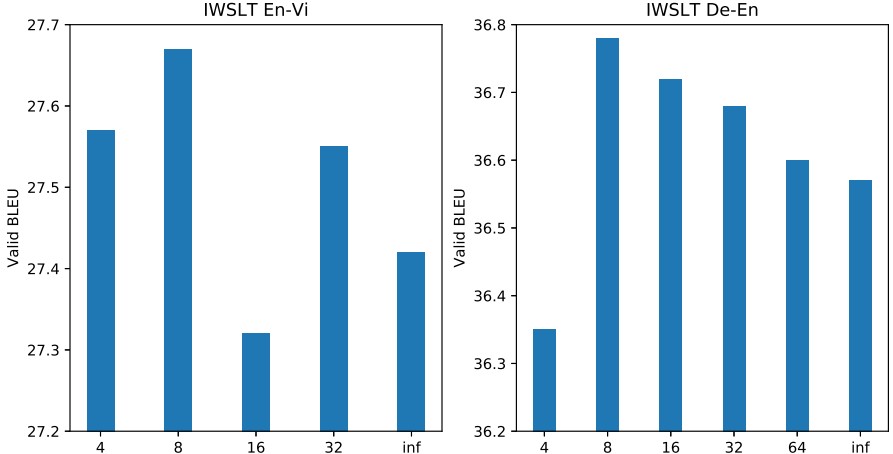

Figure 3: Analyse the value of K on IWSLT En-Vi and De-En datasets. "inf" denotes the special case of the Explicit Sparse Transformer where all positions may be attended, same as the origin Transformer.

| Task | Base | T | T&P |
|---|---|---|---|
| En-Vi (BLEU) | 27.4 | 27.7 | 27.8 |

Table 5: Results of the ablation study of the sparsification at different phases on the En-Vi test set. "Base" denotes vanilla Transformer. "T" denotes only adding the sparsification in the training phase, and "T&P" denotes adding it at both phases as the implementation of Explicit Sparse Transformer does.

### 5.2 HOW TO SELECT A PROPER K?

The natural question of how to choose the optimal $k$ comes with the proposed method. We compare the effect of the value of $k$ at exponential scales. We perform experiments on En-Vi and De-En from 3 different initializations for each value of $K$, and report the mean BLEU scores on the valid set. The figure 3 shows that regardless of the value of 16 on the En-Vi dataset, the model performance generally rises first and then falls as $k$ increases. Under the setting of the $k \in \{4, 8, 16, 32\}$, setting the value of $k$ to 8 achieves consistent improvements over the

### 5.3 DO THE PROPOSED SPARSE ATTENTION METHOD HELPS TRAINING?

We are surprised to find that only adding the sparsification in the training phase can also bring an improvement in the performance. We experiment this idea on IWSLT En-Vi and report the results on the valid set in Table 5, . The improvement of 0.3 BLEU scores shows that vanilla Transformer may be overparameterized and the sparsification encourages the simplification of the model.

### 5.4 DO THE EXPLICIT SPARSE TRANSFORMER ATTEND BETTER?

To perform a thorough evaluation of our Explicit Sparse Transformer, we conducted a case study and visualize the attention distributions of our model and the baseline for further comparison. Specifically, we conducted the analysis on the test set of En-Vi, and randomly selected a sample pair of attention visualization of both models.

The visualization of the context attention of the decoder's bottom layer in Figure 4(a). The attention distribution of the left figure is fairly disperse. On the contrary, the right figure shows that the sparse attention can choose to focus only on several positions so that the model can be forced to stay focused. For example, when generating the phrase "for thinking about my heart"(Word-to-word translation

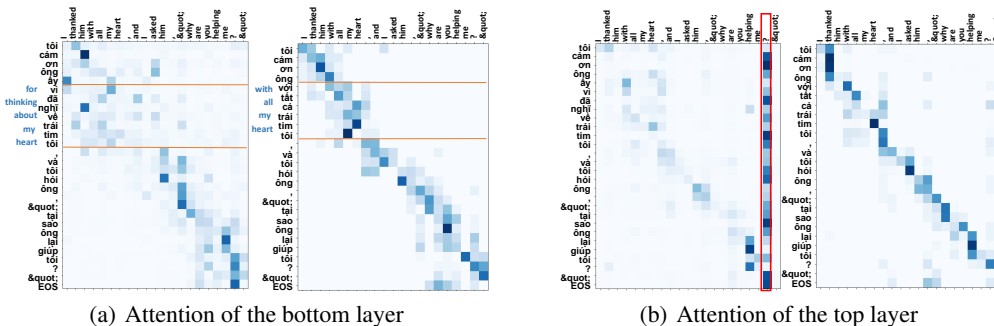

(a) Attention of the bottom layer     (b) Attention of the top layer

Figure 4: Figure 4(a) is the attention visualization of Transformer and Figure 4(b) is that of the Explicit Sparse Transformer. The red box shows that the attentions in vanilla Transformer at most steps are concentrated on the last token of the context.

from Vietnamese), the generated word cannot be aligned to the corresponding words. As to Explicit Sparse Transformer, when generating the phrase "with all my heart", the attention can focus on the corresponding positions with strong confidence.

The visualization of the decoder's top layer is shown in Figure 4(b). From the figure, the context attention at the top layer of the vanilla Transformer decoder suffers from focusing on the last source token. This is a common behavior of the attention in vanilla Transformer. Such attention with wrong alignment cannot sufficiently extract enough relevant source-side information for the generation. In contrast, Explicit Sparse Transformer, with simple modification on the vanilla version, does not suffer from this problem, but instead focuses on the relevant sections of the source context. The figure on the right demonstrating the attention distribution of Explicit Sparse Transformer shows that our proposed attention in the model is able to perform accurate alignment.

## 6 RELATED WORK

Attention mechanism has demonstrated outstanding performances in a number of neural-network-based methods, and it has been a focus in the NLP studies (Bahdanau et al., 2014). A number of studies are proposed to enhance the effects of attention mechanism (Luong et al., 2015; Vaswani et al., 2017; Ke et al., 2018). Luong et al. (2015) propose local attention and Yang et al. (2018) propose local attention for self-attention. Xu et al. (2015) propose hard attention that pays discrete attention in image captioning. Chandar et al. (2016) propose a combination soft attention with hard attention to construct hierarchical memory network. Lin et al. (2018) propose a temperature mechanism to change the softness of attention distribution. Shen et al. (2018) propose an attention which can select a small proportion for focusing. It is trained by reinforcement learning algorithms (Williams, 1992). In terms of memory networks, Rae et al. (2016) propose to sparse access memory

Child et al. (2019) recently propose to use local attention and block attention to sparsify the transformer. Our approach differs from them in that our method does not need to block sentences and still capture long distance dependencies. Besides, we demonstrate the importance of Explicit Sparse Transformer in sequence to sequence learning. Although the variants of sparsemax (Martins & Astudillo, 2016; Correia et al., 2019; Peters et al., 2019) improve in machine translation tasks, we empirically demonstrate in 5.1 that our method introduces less computation in the standard transformer and is much faster than those sparse attention methods on GPUs.

## 7 CONCLUSION

In this paper, we propose a novel model called Explicit Sparse Transformer. Explicit Sparse Transformer is able to make the attention in vanilla Transformer more concentrated on the most contributive components. Extensive experiments show that Explicit Sparse Transformer outperforms vanilla Transformer in three different NLP tasks. We conducted a series of qualitative analyses to investigate the reasons why Explicit Sparse Transformer outperforms the vanilla Transformer. Furthermore, we find

an obvious problem of the attention at the top layer of the vanilla Transformer, and Explicit Sparse Transformer can alleviate this problem effectively with improved alignment effects.

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

# A  APPENDIX

## A.1  BACKGROUND

### A.1.1  ATTENTION MECHANISM

Bahdanau et al. (2014) first introduced the attention mechanism to learn the alignment between the target-side context and the source-side context, and Luong et al. (2015) formulated several versions for local and global attention. In general, the attention mechanism maps a query and a key-value pair to an output. The attention score function and softmax normalization can turn the query $Q$ and the key $K$ into a distribution $\alpha$. Following the distribution $\alpha$, the attention mechanism computes the expectation of the value $V$ and finally generates the output $C$.

Take the original attention mechanism in NMT as an example. Both key $K \in \mathbb{R}^{n \times d}$ and value $V \in \mathbb{R}^{n \times d}$ are the sequence of output states from the encoder. Query $Q \in \mathbb{R}^{m \times d}$ is the sequence of output states from the decoder, where $m$ is the length of $Q$, $n$ is the length of $K$ and $V$, and $d$ is the dimension of the states. Thus, the attention mechanism is formulated as:

$$C = \text{softmax}(f(Q, K))V \tag{5}$$

where $f$ refers to the attention score computation.

### A.1.2  TRANSFORMER

Transformer (Vaswani et al., 2017), which is fully based on the attention mechanism, demonstrates the state-of-the-art performances in a series of natural language generation tasks. Specifically, we focus on self-attention and multi-head attention.

The ideology of self-attention is, as the name implies, the attention over the context itself. In the implementation, the query $Q$, key $K$ and value $V$ are the linear transformation of the input $x$, so that $Q = W_Q x$, $K = W_K x$ and $V = W_V x$ where $W_Q$, $W_K$ and $W_V$ are learnable parameters. Therefore, the computation can be formulated as below:

$$C = \text{softmax}\left(\frac{QK^{\text{T}}}{\sqrt{d}}\right) V \tag{6}$$

where $d$ refers to the dimension of the states.

The aforementioned mechanism can be regarded as the unihead attention. As to the multi-head attention, the attention computation is separated into $g$ heads (namely 8 for basic model and 16 for large model in the common practice). Thus multiple parts of the inputs can be computed individually. For the $i$-th head, the output can be computed as in the following formula:

$$C^{(i)} = \text{softmax}\left(\frac{Q^{(i)}K^{(i)\text{T}}}{\sqrt{d_k}}\right) V^{(i)} \tag{7}$$

where $C^{(i)}$ refers to the output of the head, $Q^{(i)}$, $K^{(i)}$ and $V^{(i)}$ are the query, key and value of the head, and $d_k$ refers to the size of each head ($d_k = d/g$). Finally, the output of each head are concatenated for the output:

$$C = [C^{(1)}, \cdots, C^{(i)}, \cdots, C^{(g)}] \tag{8}$$

In common practice, $C$ is sent through a linear transformation with weight matrix $W_c$ for the final output of multi-head attention.

However, soft attention can assign weights to a lot more words that are less relevent to the query. Therefore, in order to improve concentration in attention for effective information extraction, we study the problem of sparse attention in Transformer and propose our model Explicit Sparse Transformer.

## A.2 Experimental Details

We use the default setting in Vaswani et al. (2017) for the implementation of our proposed Explicit Sparse Transformer. The hyper parameters including beam size and training steps are tuned on the valid set.

**Neural Machine Translation**    Training For En-Vi translation, we use default scripts and hyper-parameter setting of tensor2tensor[4] v1.11.0 to preprocess, train and evaluate our model. We use the default scripts of fairseq[5] v0.6.1 to preprocess the De-En and En-De dataset. We train the model on the En-Vi dataset for $35K$ steps with batch size of $4K$. For IWSLT 2015 De-En dataset, batch size is also set to $4K$, we update the model every 4 steps and train the model for 90epochs. For WMT 2014 En-De dataset, we train the model for 72 epochs on 4 GPUs with update frequency of 32 and batch size of 3584. We train all models on a single RTX2080TI for two small IWSLT datasets and on a single machine of 4 RTX TITAN for WMT14 En-De. In order to reduce the impact of random initialization, we perform experiments with three different initializations for all models and report the highest for small datasets.

Evaluation We use case-sensitive tokenized BLEU score (Papineni et al., 2002) for the evaluation of WMT14 En-De, and we use case-insensitive BLEU for that of IWSLT 2015 En-Vi and IWSLT 2014 De-En following Lin et al. (2018). Same as Vaswani et al. (2017), compound splitting is used for WMT 14 En-De. For WMT 14 En-De and IWSLT 2014 De-En, we save checkpoints every epoch and average last 10 checkpoints every 5 epochs, We select the averaged checkpoint with best valid BLEU and report its BLEU score on the test set. For IWSLT 2015 En-Vi, we save checkpoints every 600 seconds and average last 20 checkpoints.

**Image Captioning**    We still use the default setting of Transformer for training our proposed Explicit Sparse Transformer. We report the standard automatic evaluation metrics with the help of the COCO captioning evaluation toolkit[6] (Chen et al., 2015b), which includes the commonly-used evaluation metrics, BLEU-4 Papineni et al. (2002), METEOR Denkowski & Lavie (2014), and CIDEr Vedantam et al. (2015).

**Language Models**    We follow Dai et al. (2019) and use their implementation for our Explicit Sparse Transformer. Following the previous work (Chung et al., 2015; Dai et al., 2019), we use BPC ($E[log_2 P(xt + 1|ht)]$), standing for the average number of Bits-Per-Character, for evaluation. Lower BPC refers to better performance. As to the model implementation, we implement Explicit Sparse Transformer-XL, which is based on the base version of Transformer-XL.[7] Transformer-XL is a model based on Transformer but has better capability of representing long sequences.

## A.3 The Back-propagation Process of Top-k Selection

The masking function $\mathcal{M}(\cdot, \cdot)$ is illustrated as follow:

$$\mathcal{M}(P, k)_{ij} = \begin{cases} P_{ij} & \text{if } P_{ij} \geq t_i \ (k\text{-th largest value of row } i) \\ -\infty & \text{if } P_{ij} < t_i \ (k\text{-th largest value of row } i) \end{cases} \tag{9}$$

Denote $M = \mathcal{M}(P, k)$. We regard $t_i$ as constants. When back-propagating,

$$\frac{\partial M_{ij}}{\partial P_{kl}} = 0 \quad (i \neq k \text{ or } j \neq l) \tag{10}$$

$$\frac{\partial M_{ij}}{\partial P_{ij}} = \begin{cases} 1 & \text{if } P_{ij} \geq t_i \ (k\text{-th largest value of row } i) \\ 0 & \text{if } P_{ij} < t_i \ (k\text{-th largest value of row } i) \end{cases} \tag{11}$$

---

[4]https://github.com/tensorflow/tensor2tensor

[5]https://github.com/pytorch/fairseq

[6]https://github.com/tylin/coco-caption

[7]Due to our limited resources (TPU), we did not implement the big version of Explicit Sparse Transformer-XL.

The next step after top-$k$ selection is normalization:

$$A = \text{softmax}(\mathcal{M}(P, k)) \tag{12}$$

where $A$ refers to the normalized scores. When backpropagating,

$$\frac{\partial A_{ij}}{\partial P_{kl}} = \sum_{m=1}^{l_Q} \sum_{n=1}^{l_K} \frac{\partial A_{ij}}{\partial M_{mn}} \frac{\partial M_{mn}}{\partial P_{kl}} \tag{13}$$

$$= \frac{\partial A_{ij}}{\partial M_{kl}} \frac{\partial M_{kl}}{\partial P_{kl}} \tag{14}$$

$$= \begin{cases} \dfrac{\partial A_{ij}}{\partial M_{kl}} & \text{if } P_{ij} \geq t_i \ (k\text{-th largest value of row } i) \\ 0 & \text{if } P_{ij} < t_i \ (k\text{-th largest value of row } i) \end{cases} \tag{15}$$

The softmax function is evidently differentiable, therefore, we have calculated the gradient involved in top-k selection.

## A.4 IMPLEMENTATION

Figure 5 shows the code for the idea in case of single head self-attention, the proposed method is easy to implement and plug in the successful Transformer model.

```python
import torch
import torch.nn.functional as F
def sparse_dot_product(key, value, query, k=0):
    """
            Key module of the Sparse Transformer, same time efficiency as the origin self attentio
            Compute the context vector and the attention vectors.

            Args:
                key (`FloatTensor`): set of `key_len`
                    key vectors `[batch, key_len, dim]`
                value (`FloatTensor`): set of `key_len`
                    value vectors `[batch, key_len, dim]`
                query (`FloatTensor`): set of `query_len`
                    query vectors  `[batch, query_len, dim]`
                k (`int`) select top k positions
            Returns:
                (`FloatTensor`, `FloatTensor`) :
                * output context vectors `[batch, query_len, dim]`
                * one of the attention vectors `[batch, query_len, key_len]`
    """
    # 1) Calculate self attention scores.
    scores = torch.matmul(query, key.transpose(2, 3))   # bs, query_len, key_len
    # 2) Compute the sparse attention mask and mask out the  attention values
    #    with small magnitude
    if k > key.size()[1]:
        k = key.size()[1]
    if k:
        v, _ = torch.topk(scores, k)
        # print(value)
        vk = v[:, :, -1].unsqueeze(2).expand_as(scores)
        mask_k = torch.lt(scores, vk)
        scores = scores.masked_fill(mask_k, -1e18)
    # 3) Normalize and compute context vectors.
    attn = F.softmax(scores)
    context = torch.matmul(attn, value)
    return context, attn
```

Figure 5: Code for the main idea in Pytorch

