# OpenReview forum: "Sparse Transformer: Concentrated Attention Through Explicit Selection"
_ICLR.cc/2020/Conference — Reject_

### Official Review · AnonReviewer2 · 2019-10-18
**Official Blind Review #2**

**Rating:** 1

**Review:**

CONTRIBUTIONS:
C1. Sparse Transformer: A modification of the Transformer, limiting attention to the top-k locations. (That is a complete statement of the proposed model.)
C2. Experiments showing that, quantitatively, the Sparse Transformer out-performs the standard Transformer on translation, language modeling, and image captioning.
C3. Experiments showing that, qualitatively, in translation, when generating a target word, the Sparse Transformer better focusses attention on the aligned source word

RATING: Reject

REASONS FOR RATING (SUMMARY). The innovativeness seems low given the several previous proposals for sparse attention, the results are not dramatic enough to compensate for the lack of originality, and the comparison to other models is wanting.

REVIEW

Strengths: The paper is clearly written. The question of whether the Transformer’s attention is too diffuse is of interest. The proposal is admirably simple. The quantitative metrics include comparison against many alternative models.

Weaknesses: A primary area of deficiency concerns the relation of the proposed model to other proposals for sparse attention: the authors cite 5 of them (and 2 more are cited in the comment by Cui). The paper should clearly identify the differences between the proposed model and earlier models: it does not discuss this at all. The deficiencies in these previous models should be clearly stated and demonstrated:  they are only described as “either restricted range of attention or training difficulty” (Sec 6). A rationale for why the proposal can be expected to remedy these deficiencies should be stated clearly: it is not stated at all. Experimental demonstration that the proposed innovation actually remedies the identified deficiencies should be provided, but is not.

A proposal to use a top-k filter immediately raises the question of the value of k. This is not discussed at all. In particular, no empirical results are given concerning the sensitivity of the reported successes to choosing the correct value for k. We are only told that “k is usually a small number such as 5 or 10” (Sec 3). The experimental details in the appendix do not even state the value of k used in the models reported.

It is an interesting discovery that in the translation task, attention at the top layer of the standard Transformer is strongly focused on the end of the input. This is described as an “obvious problem” (Sec 7). But it can’t obviously be a problem because the performance of the standard Transformer is only very slightly lower than that of the Sparse Transformer: if anything is obvious, it is that processing in the standard Transformer packs a lot of information into its final encoding of the end of the input string, which functions rather like an encoding of the entire sentence.

Presumably, the experimental results reported are those from a single model, since we are not told otherwise. There should be multiple tests of the models with different random initializations, with the means and variances of measures reported. It is possible, however, that limitations of computational resources made that infeasible, although the Appendix seems to indicate that no hyperparameter tuning was done, which greatly reduces computational cost.

COMMENTS FOR IMPROVEMENT, NOT RELEVANT TO RATING DECISION

Although the tiny sample of visualized attention weights provided is useful, a large-scale quantitative assessment of a main claim concerning translation might well be possible: that attention is in fact concentrated on the aligned word might be testable using an aligned bilingual corpus or perhaps an existing forced aligner could be used.

Much space could be saved: it is not necessary to review the standard Transformer, and the modification proposed is so simple that it can be precisely stated in one sentence (see C1 above): the entire page taken up by Sec. 3 is unnecessary, as it adds only implementation details.

Errors that took more than a moment to mentally correct, all on p. 12:

The definition of the BPC should be E[log P(x(t+1) | h(t))]: all parentheses are missing
“regrad” should be “regard”
“derivative” should be “differentiable” in the final sentence

**Experience Assessment:**

I have published in this field for several years.

**Review Assessment: Checking Correctness Of Derivations And Theory:**

I assessed the sensibility of the derivations and theory.

**Review Assessment: Checking Correctness Of Experiments:**

I assessed the sensibility of the experiments.

**Review Assessment: Thoroughness In Paper Reading:**

I read the paper thoroughly.

---

> ### Comment · AnonReviewer2 · 2019-11-15
> **AnonReviewer2 Response**
>
> Having read the other reviews and the limited discussion, I feel that the summary of the reasons given for the rating in my original review still stands, so my rating remains unchanged, 1: Reject. The other reviews and the responses to them reinforced my original decision to reject the paper.

---

> > ### Author Response · Authors · 2019-11-15
> > **Sorry for the late reply and updates**
> >
> > The response and paper updates may answer some of your concerns.

---

> ### Author Response · Authors · 2019-11-15
> **Response to Reviewer2**
>
> Thank you for your detailed and helpful reviewers.
> We have updated the pdf.
> Questions about the novelty and significance:
>
> In the paper Explicit sparse Transformer, the proposed method is straightforward, simple, and easy to implement.
>
> We invested a lot of time in the study of sparse transformers. In January of this year, we submitted a model to SQuAD in the name of Sparse Transformer (but it doesn't work because we do not apply the mothod to the pretrain  phase at the time). We also thought about other complicated methods of sparsee attention, but the current method is simple and effective.
>
> Although several sparse attention methods have been applied to the sequence-to-sequence transformer model and improve the performance, detailed comparisons between the proposal and their methods based on strong baseline show that our methods are much faster in training and testing and achieve slightly better results
>
> Question about the number of experiments
> In the current version of the paper, for all methods on IWSLT datasets, we have experimented under three different initializations, and reported the highest results. Because of resource limits, we didn’t do this on other data sets.
>
> Question about the Alignment of Transformer:
> We found that the randomly selected samples of the last two layers of transformer would cause excessive attention to the end token.
>
> Question about the redundancy:
> We have moved the review of standard transformer into Appendix and it may help newcomers.

---

### Official Review · AnonReviewer3 · 2019-10-21
**Official Blind Review #3**

**Rating:** 3

**Review:**

1. What is the specific question/problem tackled by the paper?

The authors tackle the problem of sparse attention for various generative modeling tasks such as machine translation and image captioning. The main motivation behind studying this problem is the premise that sparse varieties of attention might generalize better than full attention. The authors propose a sparse attention mechanism based on the top-k selection where all attention values in a row are dropped if they are not higher than the k^{th} largest item in the row. Since this is a non-differentiable operation the authors propose to train this model by setting the gradients of the non-selected items to 0. The authors report results on machine translation, language modeling and image captioning.

2. Is the approach well motivated, including being well-placed in the literature?

In my view the main reasons to study sparse variants of attention are either 1) scale to sequences longer than are possible with full attention (this is e.g., the motivation behind [1]) or 2) generalize better than full attention. The motivation of this work seems to be the latter as the authors claim improvements in terms of performance over full attention. The authors cite prior work on sparse attention mechanisms.

3. Does the paper support the claims? This includes determining if results, whether theoretical or empirical, are correct and if they are scientifically rigorous.

The authors report good results on machine translation, showing that their sparse attention method improves performance on En-De to 29.4 BLEU, on De-En to 35.6 BLEU and on En-Vi to 31.1 BLEU, improving on full attention baselines. However, the authors have not submitted code for reproducing their results. The authors also do not report what choice of k is used for the top-k operation and how they made their choice of the optimal k? The paper would be well served by more ablation experiments demonstrating what the impact the choice of k has on the model performance. For example, I would expect to be able to reproduce original Transformer results using k = maximum sequence length.

I am also not fully clear about how gradients are propagated through the top-k operation. It seems that if an index is not selected (i.e. it's attention value is smaller than top-k) it's gradient is set to 0. However, this seems problematic - for e.g., in the initial stages an important item might have a low attention value due to random initialization and might not make it to the top-k. Because of the way gradients are propagated it will not receive any gradient, and therefore will not be incentivized to increase its value. This doesn't seem like a good solution to me.

Since the paper is mainly an empirical work, it would be improved by open-sourcing anonymized code so that it's results and claims may be verified. It would also be improved in more ablation experiments or explanations in what the optimal choice of k should be for the top-k and how that affects the results.

[1] Generating Long Sequences with Sparse Transformers by Child et al (https://arxiv.org/abs/1904.10509)



**Experience Assessment:**

I have published one or two papers in this area.

**Review Assessment: Checking Correctness Of Derivations And Theory:**

I carefully checked the derivations and theory.

**Review Assessment: Checking Correctness Of Experiments:**

I carefully checked the experiments.

**Review Assessment: Thoroughness In Paper Reading:**

I read the paper thoroughly.

---

> ### Author Response · Authors · 2019-11-15
> **Response to reviewer3**
>
> Thank you for your valuable comments. We have empirically addressed your concerns about the optimal choice of k and the comparisons to the previous sparse attention methods in the updates.
>
> As you said, our approach is simple, so the Explicit Sparse Transformer is significantly faster in both inference and training than the previous methods of sparse attention in Transformer.
>
> For open-sourcing, we provide a simple implementation of the method in the Appendix, and we will publicly release all the code and training instructions in the near future to help replicate this work.
>
> For sparse attention methods of local attention(OpenAI’s sparse transfomers, adaptive span), these methods directly ignore long-distance dependence, and they mainly work for language models but have not been proved effective on standard transformers. Therefore, we did not compare them in the experiment, but we take the variants of sparsemax into consideration because they have demonstrate improvement in standard transformer.

---

### Official Review · AnonReviewer1 · 2019-10-23
**Official Blind Review #1**

**Rating:** 6

**Review:**

The paper proposes "sparse self-attention", where only top K activations are kept in the softmax. The resulting transformer model is applied to NMT, image caption generation and language modeling, where it outperformed a vanilla Transformer model.

In general, the idea is quite simple and easy to implement. It doesn't add any computational or memory cost. The paper is well written and easy to read. The diverse experimental results show that it brings an improvement. And I think this can be combined with other improvements of Transformer.

However, there are quite many baselines are missing from the tables. The sota on De-En is actually 35.7 by Fonollosa et.al. On enwik8, Transformer XL is not the best medium sized model as the authors claimed. See below:

NTM En-De:
- Wu et.al. Pay Less Attention with Lightweight and Dynamic Convolutions, 2019
- Ott et.al. Scaling Neural Machine Translation, 2018
NTM En-Vi:
- Wang et.al. SwitchOut: an Efficient Data Augmentation Algorithm for Neural Machine Translation, 2018
NTM De-En:
- Wu et.al. Pay Less Attention with Lightweight and Dynamic Convolutions, 2019
- Fonollosa et.al. Joint Source-Target Self Attention with Locality Constraints, 2019
- He et.al. Layer-Wise Coordination between Encoder and Decoder for Neural Machine Translation, 2018
LM Enwik8:
- Sukhbaatar et.al, Adaptive Attention Span in Transformers, 2019

Other comments:
- More experimental details are needed. What is the value K? How different K values affect performance? What is the number of parameters of NMT models.
- The claim "top layer of the vanilla Transformer focuses on the end position of the text" can't be true generally. Probably only true for a certain task.
- Where the numbers in Figure 1 come from? Is it a single attention head or average of all?
- Page 4, "the high are ..." probably typo?
- The related work is missing "Scaling Memory-Augmented Neural Networks with Sparse Reads and Writes" by Rae et.al., which also uses sparse attention.

**Experience Assessment:**

I have published one or two papers in this area.

**Review Assessment: Checking Correctness Of Derivations And Theory:**

N/A

**Review Assessment: Checking Correctness Of Experiments:**

I carefully checked the experiments.

**Review Assessment: Thoroughness In Paper Reading:**

I read the paper thoroughly.

---

> ### Author Response · Authors · 2019-11-15
> **Response to Reviewer1**
>
> Thanks for your careful reviews, analysis of the value of k, comparisons between previous sparse attention methods, and missing reference are all included in the newer version of the paper. The answers to the remaining questions are as follows:
>
> Q: More experimental details are needed. What is the number of parameters of NMT models.
> A: We have added more experimental details to the Appendix. Since our model does not increase the number of parameters, the parameter quantities of the machine translation model are the same as the Transformer base and the Transformer large respectively on the IWSLT and WMT datasets.
>
> Q: The claim "top layer of the vanilla Transformer focuses on the end position of the text" can't be true generally. Probably only true for a certain task.
> A: we observe similar phenomena in other tasks, such as IWSLT German to English translation.
>
> Q: Where the numbers in Figure 1 come from? Is it a single attention head or average of all?
> A: For the sake of simplicity, we show the attention score of first head.

---

### Public Comment · ~Hao_Zhang1 · 2019-10-10
**Questions about your paper**

I have some questions about your work:

1) You said your proposed sparse attention can extend to context attention. You use W_Qs to replace Q. Question: what does decoding states s means? could you give some examples?

2) How to perform image caption by Transformer? Maybe you can give clearly illustration about that?

3) Maybe you can give some visualization of image caption results?

---

> ### Author Response · Authors · 2019-10-12
> **Answer your questions**
>
> To Q1: Since we apply teacher forcing, we feed the shift right ground truth in the training phase or the generated words in the valid or test phase into the decoder, and decoding states s means the c representation
>
> To Q2: We use pertained ResNet to exact feature maps, and then feed these feature maps into the encoder and then formulate it as a sequence-to-sequence task.
>
> To Q3: Thanks for your advice. The image caption is similar to machine translation, and due to length limit, we do not include its visualization yet.

---

### Public Comment · ~Hongyi_Cui1 · 2019-10-17
**Comparision with some related work.**

Thanks for the impressive work. Top-k method seems to be very effective and easy to implement.
Have you compared with some related work with sparsemax function such as [1, 2] ?  Concentrate attention berfor softmax or during softmax, which one would be better？

[1]Chaitanya Malaviya, Pedro Ferreira, André F. T. Martins. Sparse and Constrained Attention for Neural Machine Translation. ACL 2018
[2]Gonçalo M. Correia, Vlad Niculae, André F.T. Martins. Adaptively Sparse Transformers. EMNLP 2019

---

> ### Author Response · Authors · 2019-11-03
> **Results of sparsemax, entmax-1.5, entmax-alpha**
>
> Hi，we test the above 3 sparsemax variants   on fairseq platform, envi and deen translation datasets woth single 1080ti and FP32 training. For speed, sparse transformer is 160k tokens per second, entmax-1.5 is 150k, sparsemax is  140+k and entmax-alpha trains with only 80k tokens per second.
>
> For  results, sparsemax converges  much slowly and get much worse results. Entmax-1.5  get 0.1 BLEU scores better than the implemented transformer baseline on both two datasets, and entmax-alpha get 0.2 BLEU scores worse than the baseline, if I use the entmax correctly.

---

### Public Comment · ~Wenjie_Li3 · 2019-11-08
**question about Top K operation.**

It's very interesting that the top k self-attention performs such well in those tasks listed in paper. since the top-k operation is not differentiable, the implementation/approximation method plays an important role.
do you plan to publish your implementation code of top-k attention?

---

### Author Response · Authors · 2019-11-15
**General Response and we have updated the pdf to address most of the reviewers' questions.**

We not only analyze the value of k but also compare our methods with previous sparse attention methods in transformer in the revision. In these new experiments, we performed experiments for each method under three different initializations. Our method has achieved slightly better results than the previous sparse attention method， but the inference and training speed are much faster than the previous methods. For example, in the transformer model, our method is twice as fast as the sparsemax during the inference. We empirically tried different values of k on the valid set of two translation datasets. As the value of k increases, the BLEU scores rises first and then falls, and the optimal k is around 8.

---

### Decision · Program_Chairs · 2019-12-19

**Decision:**

Reject

**Comment:**

The paper proposes a variant of Sparse Transformer where only top K activations are kept in the softmax. The resulting transformer model is applied to NMT, image caption generation and language modeling, where it outperformed a vanilla Transformer.

While the proposed idea is simple, easy to implement, and it does not add additional computational or memory cost, the reviewers raised several concerns in the discussion phase, including: several baselines missing from the tables; incomplete experimental details; incorrect/misleading selection of best performing model in tables of results (e.g. In Table 1, the authors boldface their results on En-De (29.4) and De-En (35.6) but in fact, the best performance on these is achieved by competing models, respectively 29.7 and 35.7. The caption claims their model "achieves the state-of-the-art performances in En-Vi and De-En" but this is not true for De-En (albeit by 0.1). In Table 3, they boldface their result of 1.05 but the best result is 1.02; the text says their model beats the Transf-XL "with an advantage" (of 0.01) but do not point out that the advantage of Adaptive-span over their model is 3 times as large (0.03)).

This prevents me from recommending acceptance of this paper in its current form. I strongly encourage the authors to address these concerns in a future submission.